# Impact of Cigarette Smoking on the Expression of Oxidative Stress-Related Genes in Cumulus Cells Retrieved from Healthy Women Undergoing IVF

**DOI:** 10.3390/ijms222313147

**Published:** 2021-12-05

**Authors:** Fani Konstantinidou, Maria Cristina Budani, Annalina Sarra, Liborio Stuppia, Gian Mario Tiboni, Valentina Gatta

**Affiliations:** 1School of Medicine and Health Sciences, “G. d’Annunzio” University of Chieti-Pescara, 66100 Chieti, Italy; fanikonst@hotmail.com (F.K.); stuppia@unich.it (L.S.); 2Unit of Molecular Genetics, Center for Advanced Studies and Technology (CAST), “G. d’Annunzio” University of Chieti-Pescara, 66100 Chieti, Italy; 3Department of Medical, Oral and Biotechnological Sciences, “G. d’Annunzio” University of Chieti-Pescara, 66100 Chieti, Italy; maria.budani@unich.it; 4Department of Philosophical, Pedagogical and Quantitative Economic Sciences, “G. d’Annunzio” University of Chieti-Pescara, 66100 Chieti, Italy; annalina.sarra@unich.it

**Keywords:** cigarette smoke, cumulus cells, oxidative stress, gene expression, IVF, female reproduction, infertility, epigenetics, DNA methylation

## Abstract

The female reproductive system represents a sensitive target of the harmful effects of cigarette smoke, with folliculogenesis as one of the ovarian processes most affected by this exposure. The aim of this study was to analyze the impact of tobacco smoking on expression of oxidative stress-related genes in cumulus cells (CCs) from smoking and non-smoking women undergoing IVF techniques. Real time PCR technology was used to analyze the gene expression profile of 88 oxidative stress genes enclosed in a 96-well plate array. Statistical significance was assessed by one-way ANOVA. The biological functions and networks/pathways of modulated genes were evidenced by ingenuity pathway analysis software. Promoter methylation analysis was performed by pyrosequencing. Our results showed a down-regulation of 24 genes and an up-regulation of 2 genes (IL6 and SOD2, respectively) involved in defense against oxidative damage, cell cycle regulation, as well as inflammation in CCs from smoking women. IL-6 lower promoter methylation was found in CCs of the smokers group. In conclusion, the disclosed overall downregulation suggests an oxidant-antioxidant imbalance in CCs triggered by cigarette smoking exposure. This evidence adds a piece to the puzzle of the molecular basis of female reproduction and could help underlay the importance of antioxidant treatments for smoking women undergoing IVF protocols.

## 1. Introduction

The habit of smoking is widespread. In 2019, nearly 14 of every 100 U.S. adults aged 18 years or older (14.0%) smoked cigarettes [1]. Among the 4000 chemicals that constitute cigarette smoke (CS), the polycyclic aromatic hydrocarbons (PAHs), nitrosamines, heavy metals, alkaloids and aromatic amines represent the most studied classes of toxicants [2]. The male and female reproductive systems represent sensitive targets of the harmful effects of cigarette smoke [3,4,5]. In women, cigarette smoke exposure is associated with earlier menopause and delayed conception [6]. In addition, cigarette smoking is linked to lower clinical pregnancy and live birth rates, together with increased spontaneous miscarriage rate [7,8,9,10,11], both in couples naturally conceiving and in those undergoing in vitro fertilization (IVF) techniques [6,7] compared to non-smoking couples. The ovaries are highly vulnerable to the deranging effects of cigarette smoking; steroidogenesis and folliculogenesis are the ovarian processes most affected by this exposure [4,5]. In addition, the cigarette smoke-induced oxidative stress (OS) appears to be one of the main causes of ovarian injury [12,13,14], together with the abnormal crosstalk between oocyte and granulosa-cells, DNA damage and increased cell death [2,4,5,15]. The OS is defined as the dysregulation between the endogenous antioxidant defense mechanisms and the production of reactive oxygen species (ROS), with a consequent excess of ROS [12]. The presence of ROS at low concentrations exerts a physiological role for cell homeostasis. Adequate amounts of ROS play important roles in the ovary, with particular regard for the follicles’ development and ovulation processes [16,17]. In addition, the regulation of the sperm function during the fertilization process seems to be regulated by physiological levels of ROS [18]. On the contrary, the excessive amount of ROS is responsible for cellular harm, including protein and lipid peroxidation, DNA damages, and eventually cell death. Superoxide anion (O2–) and hydrogen peroxide (H2O2) are the main species of ROS, while the enzymatic system, including the cytosolic copper–zinc superoxide dismutase (SOD1), mitochondrial manganese-SOD (SOD2), peroxidase and catalase are primarily responsible for the antioxidant defense [14]. The increased ROS formation as a consequence of cigarette smoke (or its components) exposure has been found in mouse ovarian follicles cultured in vitro [19], in oocytes [13,20] and in mice ovaries exposed to this stress [21]. Cumulus cells (CCs) originate from relatively undifferentiated granulosa cells (GCs) and surround the oocyte. The close association between oocyte and CCs allows CCs to exert central roles, supporting the maturation of the oocyte and transmitting endocrine and other environmental signals. For this reason, CCs may reflect the oocyte characteristics [22]. It has been demonstrated that cigarette smoke components affect the vitality and quality of CCs, inducing the suppression of the CCs expansion, which is a phenomenon that plays a crucial role in the processes of fertilization and embryo development [23,24]. CS-triggered inflammation is considered to play a central role in various pathologies by a mechanism that stimulates the release of pro-inflammatory cytokines. During this process, epigenetic alterations are known to play important roles in the specificity and duration of gene transcription [25]. Epigenetic research is becoming a rapidly developing and exciting area in biology. Epigenetics is usually defined as heritable alterations in gene expression patterns that are not directly caused by modifications encoded in the nucleotide genome sequence, but are caused by post-translational modifications in DNA and histone proteins and by the regulation of non-coding RNAs (ncRNAs) [26]. DNA methylation represents the major epigenetic mechanism. DNA is methylated by the transfer of a methyl group from S-adenosyl-L-methionine to covalently bind to the cytosines in CpG dinucleotides. DNA hypermethylation always leads to transcriptional suppression and decreased gene expression, while DNA hypomethylation influences chromosome stability or enhanced aneuploidy. It is very important to note that the changes caused by epigenetic modifications of DNA represent dynamic information, which indicates that they are markedly influenced by environmental stimuli, such as CS, air pollution and dietary changes [27]. The role of environmental impact on CCs’ epigenetic profile modification and the development of competent oocytes is known [28]. The majority of this information derived from experimental studies conducted in vitro and in vivo using animal models. Little data is available regarding the effects of cigarette smoke exposure on the expression of oxidative stress-related genes in human CCs, together with the epigenetic regulation behind cigarette smoke-mediated ovotoxicity. In this context, the DNA methylation represents a key epigenetic process, which is crucial to the regulation of gene expression [29]. To the best of our knowledge, the evaluation of CCs gene profile in smoking versus non-smoking women has not been previously carried out. Evidence that cumulus cells have a central role in the development of oocyte competence, alongside available information about noxious effects of cigarette smoking on fertility, led us to investigate the impact of tobacco smoking on modulation of oxidative stress-related gene expression in CCs of smokers undergoing IVF techniques. To this aim, real time PCR technology was used to analyze the gene expression profile of oxidative stress genes enclosed in 96-well plate arrays, as well as pyrosequencing target gene approaches for promoter methylation status.

## 2. Results

### 2.1. Patients’ Characteristics

We collected CCs from ten donors: five smokers and five non-smokers. Patients’ demographic characteristics were reassumed in Table 1. No statistically significant differences exist among the two populations of the study.

### 2.2. Gene Expression

Statistical analysis on a total of 88 oxidative stress-related genes showed a down-regulation of 24 genes (*NFKB1 h*, *APEX1*, *BCL2*, *CAT*, *CYBA*, *EGFR*, *F3*, *FOXO1*, *GCLC*, *HIF1A*, *IL1B*, *JUN*, *MAP3K5*, *MAPK1*, *MAPK8*, *NFE2L2*, *NQO1*, *SIRT1*, *GPX3*, *HMOX1*, *MSRA*, *SERPINE.1*, *STAT1* and *TP53*) (*p* < 0.05) involved, overall, in cell repair and defense against oxidative damage, as well as in the regulation of apoptosis and proliferation, in CCs of all or the majority of smoking women compared to their corresponding age-matched controls. A significant up-regulation of 2 genes, interleukin 6 (IL6) (*p* = 0.007) and superoxide dismutase 2 (SOD2) (*p* = 0.013), was also detected in the majority of smokers indicating the presence of inflammation due to oxidative stress and a consequent protective response to damage (Figure 1 and Figure 2).

### 2.3. Functional and Network Analysis

IPA analysis, performed to highlight the main functions and cellular processes in which these genes are involved, found that they were mainly involved in the following cellular functions: cell death and survival, cell cycle, cell signaling, cellular growth and proliferation, gene expression, free radical scavenging, cellular function and maintenance and DNA replication, recombination and repair (Figure 3).

These genes are also mainly located in the nucleus, cytoplasm, plasma membrane and in the extracellular space; they are mostly transcription regulators, and are also involved in cytokine (IL1B, IL6) and kinase (MAP3K5, MAPK1, MAPK8) activity (Table 2).

IPA analysis was also used to highlight the networks in which these genes are involved. Three networks, with a score ranging from 39 to 8 (data not shown), were indicated. The top network generated by IPA is provided in Figure 4 and is centered around the key node gene IL-6 gene, which actively participates in the inflammatory response. The top pathway inferred by IPA is represented by the NRF2 signaling pathway (Figure 5). The NRF2 signaling pathway is one of the most relevant defense strategies against oxidative stress. Under normal physiological conditions, Nrf2 ubiquitination leads to the Nrf2 accumulation. Consequently, Nrf2 translocates into the nucleus, where it targets a plethora of genes playing important roles in the survival of the cells. The products of such genes are vastly different according to their function. They include proteins involved in the antioxidant response-detoxification process, enzymes involved in cell survival and tumorigenesis and various mediators of protein repair and removal as shown in Figure 5 (for an extensive review, see Shaw P, et al., 2020) [30]. Pathway activity analysis predicted, based on the expression of several significantly-perturbed genes from our dataset, the inhibition of this pathway (blue shape lines in Figure 5), suggesting an obstruction in the reduction of oxidative damage and cell survival.

NRF2, circled in pink, represents the master regulator as a pro-survival factor that regulates the production of cytoprotective machinery components under adverse conditions. Blue shapes predict a general pathway inhibition. Predictions were calculated by the experimental dataset overlaid onto the ingenuity knowledge base in IPA.

### 2.4. IL-6 Promoter Methylation through Pyrosequencing

Pyrosequencing performed on the cumulus cells retrieved from healthy female smokers compared to their corresponding controls (non-smokers) undergoing IVF showed a significant methylation alteration regarding the majorly up-regulated gene, interleukin 6 (IL-6). More specifically, average methylation of the two sites studied was estimated at 41% in smokers vs. 50% in non-smokers (*p* ≈ 0.027) indicating a lower methylation percentage of IL-6 in CCs of the first ones and, consequently, supporting its previous up-regulation in terms of gene expression (Figure 6, Appendix A Appendix A).

## 3. Discussion

The cumulus cell-oocyte communication is essential for the correct oocyte development. The bidirectional signaling through gap junctions between oocytes and surrounding cumulus cells regulates several aspects of the oocyte health, including oocyte developmental competence and oocyte meiotic maturation [31]. The exposure to tobacco smoke constituents is known to potentially harm human reproductive health, with particular reference to the ovary. Several experimental studies conducted in vitro or in vivo showed that nicotine (with its principal metabolite-cotinine), together with benzo[a]pirene (BaP), dimethylbenz[a]anthracene (DMBA) and cadmium, exert central roles in the process of ovotoxicity, influencing the steroidogenesis, the correct follicle development and the correct meiotic progression of the oocytes [2,4,5]. Although a number of studies have explored the mechanisms behind cigarette smoke-induced ovarian toxicity, the understanding of this process remains, so far, not completely elucidated. For these reasons, this study aimed to evaluate for the first time the impact of cigarette smoking on the expression of oxidative stress-related genes in human cumulus cells of female smokers undergoing IVF techniques compared to age-matched non-smoking controls. The experiments evidenced a general down-expression profile, supported by the significant down regulation of 24 genes, as well as the up-regulation of two genes in the smokers’ group versus the non-smokers’ one. The down-regulated genes, as disclosed by IPA functional analysis, have been shown to be linked to cell repair and survival, control of the cell cycle and apoptosis as well as oxidative stress regulators, all of which could be potentially central to consequent impaired follicular development following exposure to cigarette smoke. In brief, the cellular antioxidant system balances generation and scavenging of ROS to maintain normal cellular functions. However, excessive ROS production beyond antioxidative capacity may result in a condition called oxidative stress. The data presented in this study provides evidence that cumulus cells from female smokers show a general down-regulation of genes, mainly related to cellular antioxidant molecules, compared to non-smokers (*APEX1*, *CAT*, *CYBA GCLC*, *MSRA NFE2L2*, *NQO1*, *SIRT1*, *GPX3*), which may contribute to a stress-related increase in cellular oxidative damage in the ovary. In addition, the down-regulation of genes related to regulation of the cell cycle was also observed (*NFKB1 h*, *BCL2*, *EGFR*, *F3*, *FOXO1*, *HIF1A*, *IL1B*, *JUN*, *MAP3K5*, *MAPK1*, *MAPK8*, *HMOX1*, *SERPINE1*, *STAT1* and *TP53*). The overall results support the idea that cigarette smoking impairs the health of cumulus cells by modulating the expression of genes involved in a protective cellular response to oxidative stress-caused damages. With respect to our data, the evidenced de-regulation of gene expression could possibly be associated with a decline of CCs in terms of vitality and proliferation, not being able to mitigate the noxious effects of cigarette smoking. The *CAT* gene, for instance, encodes catalase, an antioxidant enzyme which plays a key role in the organism’s defense against oxidative stress. Catalase converts hydrogen peroxide to water and oxygen and by doing so it manages to avoid its toxic effects. Expression levels of *CAT* have been also shown to be lower in cultured granulosa cells collected from 38- to 42-year-old patients undergoing in vitro fertilization, (IVF) compared to those collected from 27- to 32-year-old IVF patients; in addition, they have been associated with ultrastructural modifications in human granulosa cells [32]. The *MSRA* gene encodes a protein, which is highly conserved and responsible for carrying out the enzymatic reduction of methionine sulfoxide to methionine. The protein in question repairs oxidatively damaged proteins and reinstates biological activity. The protection offered by MsrA against oxidative stress has been described in many cell models such as cardiac myocytes, cultured mouse embryonic stem cells and WI-38 SV40 human fibroblasts. Furthermore, lifespan of *MsrA* knockout mice has been found to be shorter, which may be the result of oxidized proteins’ accumulation [33]. The protein encoded by the *GPX3* gene is a member of the glutathione peroxidase family, which catalyzes the reduction of organic hydroperoxides by glutathione, and, therefore, shields cells against oxidative damage. Performed microarray analysis has indicated GPX3 as one of the most differentially down-expressed genes in non-early cleavage-CCs samples, probably delaying oocyte maturation [34]. The NFE2L2 (also known as NRF2) gene encodes a transcription factor which functions by regulating genes containing antioxidant response elements at a promoter level. A considerable amount of these genes subsequently encodes proteins involved in injury response and inflammation, including the production of free radicals. Additionally, it has been indicated that Nrf2-null mice could decrease basal and inducible expression of antioxidant genes, as well as reduce activity and antioxidant capacity, while, on the other hand, consequently increase oxidative stress. This suggests that the Nrf2/ARE pathway could be critical for the regulation of intracellular redox status [35]. In line with our data, Garbin et al. showed that the Nrf2/ARE pathway was neither activated nor repressed in peripheral mononuclear cells of heavy smokers. This was furtherly supported by in vitro findings indicating that, at the highest concentrations of PGPC, the Nrf2/ARE pathway was no longer stimulated or reduced. It has also been recently evidenced that Nrf2 could be considered a vital regulator of NF-kB activation [36]. A decline of Nrf2 in granulosa cells may cause Nrf2 levels inside the oocytes to also decrease, resulting in a reduction in oocyte quality. This way, authors showed that expression of Nrf2 in granulosa cells could potentially decrease with age, implying that Nrf2 may be associated with decline in oocyte quality in older women, and providing an indicator of a possible clinical diagnosis [37]. *HMOX1* is a gene that encodes another antioxidant enzyme called heme oxygenase 1. This enzyme has been known to be regulated by the *NRF2* signaling pathway [38]. The antioxidative effect provided by activating *NRF2* is mainly dependent on the release of *HMOX1* [39]. Consistent with *NRF2*, it has also been evidenced that cigarette smoke-contained nicotine is able to inhibit expression of *HMOX1* by increasing oxidative stress in primary cardiomyocytes [40]. The APEX1 gene encodes the apurinic/apyrimidinic (AP) endonuclease in human cells, which is fundamental in cellular response to oxidative stress. The two principal functions of APEX1 are DNA repair and transcriptional factors’ redox regulation. In neuroblastoma cells treated with both high and low levels of hydrogen peroxide, a fairly rapid decrease in Ref-1 expression and activity associated with substantial DNA damage and degeneration of neurons has been evidenced [41]. In addition to being a repairing factor, Ape1 also acts as a major redox-signaling regulator, aiming to decrease and activate transcription factors, such as AP1, p53, HIF-1α, as well as others responsible for controlling gene expression vital for cell survival and cancer promotion and progression [42]. The CYBA gene encodes for a protein that is part of the ROS-generating NADPH oxidase complex. Mutations in this gene are associated with failure of superoxide production on behalf of activated phagocytes [43]. The pattern leading to a lower CYBA expression in smokers may be interpreted as a protective negative feedback exerted by existing tobacco smoking-related oxidative stress [44]. Sirtuin, also known as SIRT1, is a nicotinamide adenine dinucleotide (NAD+)-dependent histone deacetylase that was once considered a vital enzyme to increasing life expectancy in yeast, worms, flies and mice [45]. SIRT1 has been found to regulate many physiological and pathological processes, such as cellular senescence and aging, apoptosis, resistance to stress, autoimmunity and regulation of metabolism [46]. Scientific evidence has indicated reduced levels and activity of SIRT1 in many types of cells, both in vitro and in vivo, like in lungs of mice and patients exposed to cigarette smoke [47,48,49]. SIRT1-related deficiency debilitated mitochondrial function and increased levels of oxidative stress in the lung. Moreover, a combination of decreased nuclear NAD+ and SIRT1′s activity underlined a specific loss of mitochondrial-encoded subunits of the oxidative phosphorylation system. These results could potentially indicate that SIRT1 is crucial to maintain mitochondrial stability and to improve the status of oxidative stress. Moreover, it has been shown that signaling of the Sirt1 gene can protect murine oocytes against oxidative stress during aging [50]. Lastly, it has also been reported that there is a correlation between Sirt1 and the activation of nuclear factor-E2 related factor 2 (Nrf2) [37]. The *NQO1* gene is a member of the NAD(P)H dehydrogenase (quinone) family, a downstream target of *NRF2* and has multiple functions in cellular adaptation to stress. Silencing of Nrf2 has, specifically, indicated an impaired NQO1 antioxidant gene expression and uncontrolled ROS (H2O2) production [16]. GCLC is a subunit of an enzyme called glutamate-cysteine ligase, which catalyzes the first and rate-limiting step in glutathione (GSH) synthesis, an important cellular antioxidant. The decreased mRNA levels of GCLC have also been confirmed in advanced antral atretic follicles by immunofluorescence staining, sustaining the hypothesis that ROS levels may be increased in the granulosa cells alongside consequent inhibition of cellular growth [51]. The *BCL2* gene encodes an outer mitochondrial membrane protein that is able to obstruct the process of cellular apoptotic death. More specifically, it has been reported that ovarian granulosa cells, following treatment with ethanol for three hours, developed a significant reduction in cell viability and induced apoptosis in rat ovarian granulosa cells, possibly interfering with the transcriptional and translational regulation of anti-apoptotic *Bcl-2*, which was resultantly downregulated [52]. NFKB is a transcription regulator that is activated by a series of stimuli, both intra- and extra-cellular, such as oxidant-free radicals, cytokines, ultraviolet irradiation, and bacteria or viruses. As shown, constant inhibition of NFKB by a status of persistent oxidative stress could also lead to delayed cell growth [53]. EGFR is a protein located on the cell’s surface; it binds to the epidermal growth factor, inducing a dimerization of the receptor and autophosphorylation of tyrosine, leading to cell proliferation. Furthermore, EGFR is involved in meiotic resumption within the cumulus–oocyte complex, also known as COC. For this reason, it has been evidenced that oocyte maturation and ovulation could be potentially disrupted when expression of EGFR gets incrementally reduced, showing also a decrease in the oocytes’ ability to progress in resuming meiosis [54]. The IL1B gene encodes a protein that is a member of the interleukin 1 cytokine family. This cytokine is an important inflammatory response mediator and is involved in many cellular activities, including regulation of cellular proliferation and differentiation, follicular survival and atresia and oocyte maturation [55]. *IL1B* has, in fact, been shown to apply an immunostimulatory activity without causing, nonetheless, inflammatory effects, as well as modulate granulosa cell proliferation [56]. As reported, IL-1β contributes to control of follicle development by enabling proliferation of granulosa cells and preventing premature differentiation. This cytokine has also been found to influence apoptosis in ovarian granulosa cells [57]. IL-1 has emerged as an important factor in the age-related depletion of ovarian reserve in mice, possibly by boosting the expression of inflammatory genes and by advancing apoptotic pathways. In fact, it has been reported that interleukin (IL)-1 beta can induce the synthesis of both nitric oxide (NO) and prostaglandin (PG)E2 in dispersed ovarian cell cultures, exerting cytotoxic effects on them [58]. *MAP3K5*, *MAPK1(ERK)*, *MAPK8* function as a fundamental component of the MAP kinase signal transduction pathway. MAPK-related signaling is one of the signaling pathways that has been identified to modulate functions of granulosa cells and to support the development and maturation of oocytes. More specifically, this signaling pathway transfers extracellular stimulus signals to the cytoplasm and nucleus; by doing so, it activates a variety of transcription factors, such as Elk1 (ERK pathway) and c-Jun (JNK pathway). Moreover, it is of vital importance in the cellular response-related cascades induced by changes in the environment; moreover, it is a signaling mediator regarding the determination of cell fate, such as differentiation and survival. Likewise, MAPK mRNA could be defined as a key mediator in oocyte maturation by acting at a granulosa and CCs level. It has been evidenced that downregulation of MAPK1 in human granulosa cells may compromise their proliferation and differentiation, as well as follicular development and the production of steroid hormones [59]. Additionally, it has been observed that fibroblast cells in absence of c-Jun undergo a cell cycle arrest following exposure to UV irradiation [60]. *FOXO1* is a transcription factor involved in a constantly increasing number of physiological processes, such as apoptosis and cell cycle progression [61,62]. It is highly expressed in follicular granulosa cells during follicular growth and is known to be able to cause cell apoptosis throughout oxidative stress [63]. *FOXO1′s* decreased expression could potentially obstruct the induction of downstream proautophagic genes, which might represent a fundamental adaptive strategy on behalf of the cells in order to maintain survival of GCs against oxidative injury [64]. *HIF-1* is known as a master regulator of cellular homeostatic response following hypoxia by activating the transcription of a series of genes involved in energy metabolism, angiogenesis and apoptosis. In addition, HIF1alpha controls intercellular communication within the COCs, as well as steroidogenic activity and oocyte development rates. Inhibition of the hypoxia-inducible factor (HIF1alpha) also regulates cumulus cell function and has been found to affect bovine oocyte maturation in vitro [65]. *STAT1* is a transcription factor with a principal role in innate immunity; it serves as a potent inhibitor of growth and promoter of apoptosis. The *P53* gene encodes a tumor suppressor protein, which responds to diverse types of cellular stress in order to modulate the expression of target genes, thereby provoking arrest of the cell cycle, apoptosis, senescence, DNA repair, and changes in the metabolism. Moreover, there is evidence linking p53 and STAT-1. STAT-1 interacts directly with p53 and functions as a co-activator, regulating the functional activity of p53 responsive genes [66]. Following usage of human and murine STAT-1-deficient cells, Townsend et al. [67] indicated that STAT-1 is necessary for optimal DNA damage-induced apoptosis. STAT-1(−/−) cells, in fact, had as a consequent result a reduction in the expression of p53. Our data, in agreement with this, could support the idea of a lower p53 response to DNA damage mediated by STAT 1′s downregulation in CCs from the smokers’ group. Serpin family E member 1 (*SERPINE1*) belongs to the serine proteinase inhibitor (Serpin) superfamily. SERPINE1′s expression is restricted to CCs in rats and mice, but has been evidenced in both bovine CCs and immature and in vitro-matured oocytes in association with cumulus expansion and oocyte maturation [68]. Moreover, *Serpine1*-deficient mice were found with an increased number of apoptotic cells [69]. The superoxide dismutase (*SOD*) family has a key role in mitigating the noxious effects of ROS. Following immunohistochemistry, it has been demonstrated that an increase in SOD expression is connected to a defensive mechanism against oxidative stress. *SOD 2,* specifically, is produced in the cytoplasm and translocated, afterwards, to the mitochondrial matrix; therefore, it is able to participate in cell protection against possible damages caused by respiratory chain enzyme-generated superoxide anions [70]. *IL-6* encodes a cytokine involved in inflammation and maturation of B cells. IL-6, in fact, has been found significantly higher in smokers’ preterm preeclampsia (PPE) compared to non-smokers’ PPE, underlying a possible contribution of tobacco smoking in enhanced inflammation [71]. In CCs, it was also observed that the interleukin 6 gene could be important for cellular proliferation by being a mediator of autocrine growth [72]. It is well-known that epigenetic alterations, including methylation status, work together whilst aiming to modulate gene transcription. The enzymes responsible of regulating these epigenetic modifications can possibly be activated by smoking, which would further mediate the expression of multiple genes involved in inflammation [25]. Epigenetic dysregulation that causes a consequent inappropriate gene expression or silencing has been found to have a vital role in CS-related diseases. In this view, we analysed the promoter methylation status of IL-6 in smokers versus non- smokers, which resulted in a lower methylation percentage of IL-6 in CCs of the former. This result is in line with the gene’s up-expression observed in our experiments. When located in a gene promoter, DNA methylation typically acts to repress gene expression and, consequently, the observed lower methylation status supporting its previous up-regulation in terms of gene expression. Therefore, epigenetic changes that occur in cumulus cells might have an impact on oocyte maturation. Interleukin-6 (IL-6) is essential to inflammatory processes correlated with chronic inflammatory diseases, such as coronary heart disease (CHD). Modification of methylation is a mechanism regulating production of IL-6. Thus, it has been shown that expression of IL-6 is associated with reduced levels of its promoter’s DNA methylation [73]. Moreover, DNA methylation in IL-6′s promoter has been interlinked with other risk factors for CHD, such as exposure to air pollution. However, as far as it is known, there are no data on IL-6-related promoter methylation and the CCs’ exposure to cigarette smoke. We speculated that the hypomethylation of IL-6′s promoter may have a crucial role in upregulated inflammation cytokine IL-6 also in CCs. The exposure to cigarette smoke or its constituents altered the ovarian redox balance as evidenced by experimental data conducted in vitro or in vivo on animal models [13,19,21,74]. More to the point, a recent study of Meng et al. [75], which aimed to characterize the complex transcriptome changes in porcine granulosa cells of healthy antral (HA) and advanced antral atretic (AA) follicles, showed a general downregulation in expression of genes associated with antioxidant processes, such as GCLC, DHCR24, IDH1, TXNIP, GCLM, MSRB2, GPX8, GSTA1 and RRM2B, in the granulosa cells of AA follicles. Concerning the human counterpart, the adverse impact of smoking on female fertility is supported by clinical data. The habit of smoking cigarettes is linked with an accelerated follicular depletion that may advance the time of menopause by 1–4 years [6]. In addition, clinical evidences highlighted that smoking is negatively associated with assisted reproductive techniques (ART) outcomes [6], with a significant decrease in clinical pregnancy and live birth rates, and a significant increase in terms of spontaneous miscarriage rate for smokers [2,7]. Recently, NuÃez-Calonge et al. [76] also demonstrated that there is a lower antioxidant potential and increased levels of apoptosis in cumulus and granulosa cells of women with a low response to ovarian stimulation during ART cycles compared to women with a normal response. It has been demonstrated that smoking is associated with a depletion of the antioxidant beta-carotene [77] in human follicular fluid and with an altered expression of antioxidant enzymes in human granulosa cells [5]. Cigarette smoke contains many toxins and harmful chemicals able to potentially cause DNA damage, and cells with DNA damaged beyond repair tend to undergo apoptosis in order to safeguard their genomic integrity [78]. To this purpose, increased DNA damages have been found in human cumulus cells exposed in vitro to the toxic effects of tobacco smoke under oxidative stress conditions [79]. Under adverse conditions, CCs may be unable to defend the oocyte against alterations induced by tobacco smoke, which could, consequently, damage their functions and diminish oocyte quality. As an example of this, a study estimating the ability of ROS, such as H2O2, •OH and HOCl, to bypass the protective functions of CCs and negatively impact oocyte quality, found that CCs were able to neutralize H2O2 and •OH when found in lower concentrations, but higher concentrations of these ROS reduced both the number and vitality of CCs, ending up in a decrease in the antioxidant response, whilst making oocytes more susceptible to damage. The same study found that HOCl, no matter the concentration, negatively affected both oocytes and CCs [80]. These data suggest that when noxious factors exceed the capacity of CCs to metabolize and/or neutralize them, they may have a significant negative impact on the oocyte health, with possible influences on female fertility and the ovarian aging process.

## 4. Materials and Methods

### 4.1. Ethical Approval

The study was approved by the Institutional Review Board (Ethical Committee of Chieti and Pescara, document *n*° 05, 8 March 2018).

### 4.2. Source of Cumulus Cells and Patients’ Selection Criteria

Patients undergoing IVF techniques in the IVF unit of the General Hospital “G. Bernabeo” of Ortona (CH, Italy) donated their CCs for research purposes with a written informed consent. The patients’ inclusion criteria were: cigarette smoking habit (assessed with the use of questionnaire), age ≤39 years with regular ovulatory cycles, presence of both ovaries, normal karyotype, no previous ovarian surgery, infertility due to tubal or male factors, body mass index (BMI) of 19–25 kg/m^2^, AMH level between 0.9 e 9 ng/mL, basal FSH <15 mIU/mL, undergoing only intracytoplasmic sperm injection (ICSI) with the same protocol. The exclusion criteria were: patients with grade III or IV endometriosis according to the American Society for Reproductive Medicine (ASRM), patients with history of poor response in previous ART cycles (≤3 oocytes retrieved) or history of severe OHSS. Patients with antral follicle count (AFC) ≤7, patients with ≥3 ART attempts without a clinical pregnancy, patients with endocrinopathies and patients taking micronutrient supplements were also excluded.

Prior exposure to cigarette smoke was estimated at a mean of 10 years of total duration per woman up to the moment of oocyte retrieval.

### 4.3. Ovarian Stimulation Protocols

The controlled ovarian stimulation (COS) started on the second/third day of the menstrual cycle with an exogenous administration of 150 IU recombinant FSH (rFSH) (Gonal-f^®^, Merck Serono, Italy) or follitropin alpha biosimilars (Bemfola^®^, Gedeon Richter Italy, Milan, Italy or Ovaleap^®^, Theramex Italy, Milan, Italy). Serum 17β-estradiol measure and ovarian echography were used for monitoring the follicular growth. From stimulation day-5 the rFSH doses were adjusted on the basis of the patient’s response, until the day of human chorionic gonadotropin (hCG).

The daily injections of GnRH antagonist (ganirelix acetate) initiated in the presence of at least one follicle with an average diameter ≥14 mm, according to the flexible protocol with multi-dose injections, until the day of hCG administration (at least 3 follicles with a diameter of ≥16–18 mm). The final oocyte maturation was achieved with the administration of 250 mg/0.5 mL of choriogonadotropin alpha (Ovitrelle ^®^, Merck Serono, Milan, Italy). Oocyte–cumulus complexes (COCs) were retrieved by transvaginal ultrasound guided puncture of follicles, approximately 34–36 h after the hCG administration.

### 4.4. Cumulus Cells Isolation

The COCs retrieved at ovum pick-up were maintained in an incubator for 2 h (37 °C, 5% CO_2_ and 5% O_2_). The procedure of mechanical removal of CCs was facilitated by the exposure of COCs to hyaluronidase (HYASE-10XTM, Vitrolife, Goteborg, Sweden). At this purpose, each COC was moved into the hyaluronidase droplet (200 µL, HYASE-10X™, Vitrolife, Sweden) where it was gently blown for several times, but for no more than 30 s. At the end of this step, the oocyte in the hyaluronidase droplet was transferred into the operation droplets (50 µL) with oocyte medium (G-GAMETE, Vitrolife, Sweden), where they were gently and repeatedly aspirated with a denudation pipette. The oocytes categorized as mature (nuclear metaphase II stage) were fertilized in vitro using intracytoplasmic sperm injection (ICSI) procedure. The relative CCs were removed and washed in three droplets of medium (G-GAMETE, Vitrolife, Sweden) and were used for the aim of the present study. CCs deriving from MII oocytes were pooled and collected separately for individual patients and stored at −80 °C until the use for the genetic analysis.

### 4.5. Quantitative Real-Time PCR

Collected pools of cumulus cells were conserved in lysis buffer at −20 °C until extraction. DNA and total RNA were extracted using the Nucleospin miRNA and RNA/DNA buffer set kits (Macherey-Nagel, Milan, Italy) according to the manufacturer’s instructions. Quantity and quality of DNA and total RNA were assessed by Qubit 2.0 (Invitrogen, Monza, Italy). 1 μg of total RNA was used for cDNA synthesis performed in a 20 μL reaction through the high-capacity cDNA reverse transcription kit (Applied Biosystems, Foster City, CA, USA) under the following conditions: 25 °C for 10 min, 37 °C for 120 min, 85 °C for 5 min and cool at 4 °C. Predesigned oxidative stress 96-well plate arrays (Tier 1 H96) were used to perform quantitative real-time PCR (qRT-PCR) analysis (Bio-Rad Laboratories, CA, USA) on the ABI 7900HT sequencing detection system (Life Technologies, Carlsbad, CA, USA). Amplification reaction was carried out in a total volume of 20 μL containing 2X SsoAdvanced Universal SYBR Green Supermix (Bio-Rad Laboratories, CA, USA) and 50 ng of target cDNA. A gene was considered differentially expressed in smokers’ CCs versus control CCs when showing a fold change >1.2 or <0.7 (DataAssist Software, Thermo Scientific) using the 2−ΔΔCt method and GAPDH and HPRT1 as internal references. Statistical significance was assessed by one-way ANOVA, considering expression differences significant when *p*-value < 0.05.

### 4.6. IPA-Inferred Biological Networks and Upstream Regulators Analysis

The 26 significant de-regulated genes were analysed by ingenuity pathway analysis (IPA) software (Qiagen, Hilden, Germany). IPA-inferred network analysis was generated for the 26 transcripts linking their functionality to the function of other genes and a mechanistic network of these genes was highlighted based on their connectivity and enrichment statistics. A Fisher’s exact test was used to generate the network score, based on the number and size of eligible genes and the total number of genes that could be included in the network. An upstream effect analysis was run to predict the drivers of the expression modulation evidenced. The predicted effect is based on a value calculated by the IPA z-score algorithm. The z-score predicts the direction of change for the function (negative or positive), which was significant when showing a *p* < 0.05. The recognized up and down-regulated genes were analyzed by IPA software also in order to discover the biological functions in which they are involved. pathway activity analysis was performed to determine if canonical pathways are activated or inhibited based on the expression or phosphorylation of significantly-perturbed genes in our dataset. IPA then used the activation z-score algorithm to make predictions.

### 4.7. Bisulfite Conversion and DNA Methylation by Pyrosequencing

DNA was bisulfite-converted using the EpiTect Plus DNA bisulfite kit (Qiagen) and stored at −20 °C until utilized. PCR amplification of IL6 promoter was performed using 2x PyroMark PCR Master Mix and 10x CoralLoad Concentrate (Qiagen), 0.2 µM of each primer, 1 µL of converted-DNA and nuclease-free water to a final volume of 25 µL. Primers used for DNA methylation analysis and PCR cycling conditions are shown in Appendix A. Pyrosequencing reaction was run on a PyroMark Q96ID (Qiagen) and CpGs methylation analysis was conducted by the PyroMark CpG software (Qiagen). A triplicate was generated for each PCR. Methylation for each amplicon was calculated as the median of methylation status of each analyzed CpG. Differences in methylation pattern across samples and controls were calculated by two-tailed student T-test, considering *p*-values < 0.05 as significant.

## 5. Conclusions

In the last five decades a significant trend towards a progressive worldwide decline in human fertility has been reported in the international literature; therefore, much attention has been placed on identifying environmental and lifestyle modifiable risk factors that affect human reproductive function [81]. In the current study, several differentially expressed key molecular genes related to oxidative stress and the apoptotic pathway were identified in CCs following prolonged exposure to cigarette smoke. The overall downregulation suggests a lower antioxidant capacity in CCs of smoking versus non-smoking women. This data contributes to increase the interest around the concept that an oxidant-antioxidant imbalance could have a role in the pathogenesis of female infertility related to cigarette smoking. Reactive oxygen species can therefore trigger both apoptotic and necrotic cell death, depending on the severity of the oxidative stress. It could be maintained that, when too much cellular damage has occurred, it is helpful for a multicellular organism to remove the damage-causing cell for the benefit of the surrounding cells. However, the relationship between some antioxidants and apoptotic markers in CCs and the response to ovarian stimulation needs to be evidenced with functional and larger-scale studies. This evidence contributes to furtherly comprehend the molecular basis of female reproduction and consider the importance of antioxidant treatments for smoking women undergoing IVF protocols.

## Figures and Tables

**Figure 1 ijms-22-13147-f001:**
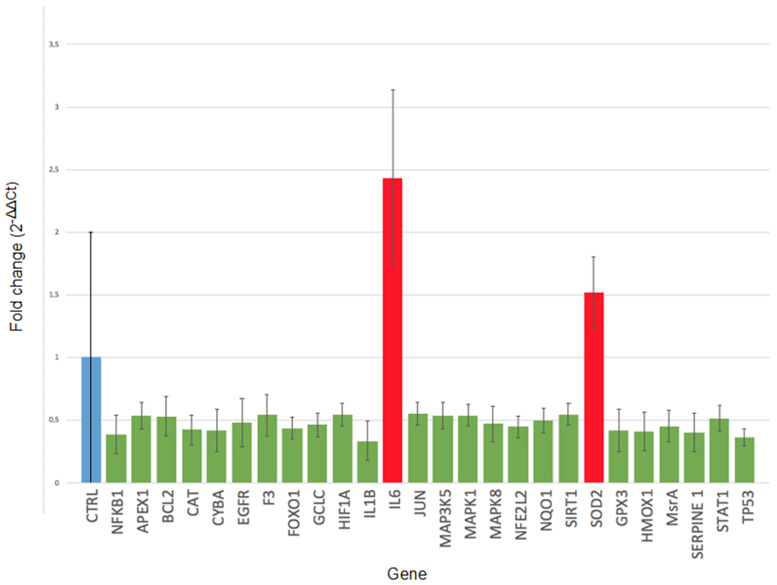
Significant mean fold change values for all differentially modulated genes in cumulus cells of female smokers undergoing IVF treatments (*p* < 0.05).

**Figure 2 ijms-22-13147-f002:**
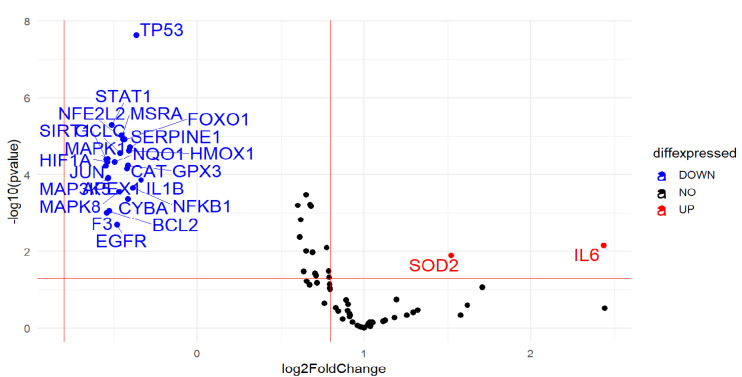
Volcano plot of statistical significance against fold- change of differentially expressed genes between female smokers and non-smokers indicating the most upregulated genes towards the right, the most downregulated genes towards the left and the most statistically significant genes towards the top.

**Figure 3 ijms-22-13147-f003:**
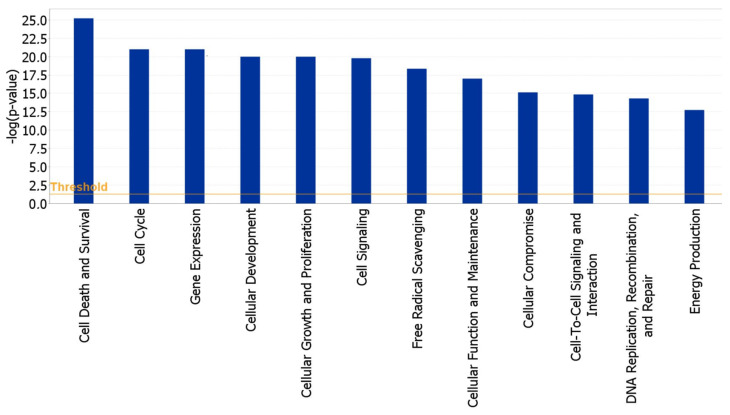
Ingenuity pathway analysis (IPA)-generated bar-chart indicating the main significant biological functions regulated by our gene dataset in cumulus cells.

**Figure 4 ijms-22-13147-f004:**
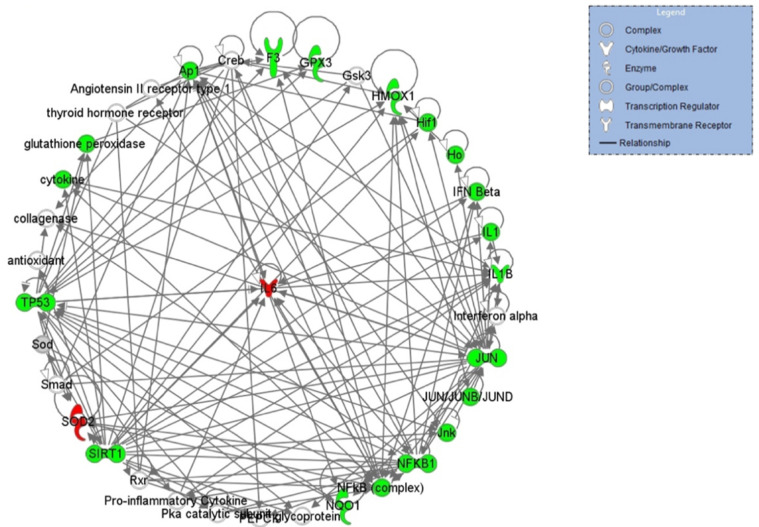
IPA-inferred target genes network for oxidative stress in female smokers’ cumulus cells. In red the up-regulated genes, while in green the down-regulated ones.

**Figure 5 ijms-22-13147-f005:**
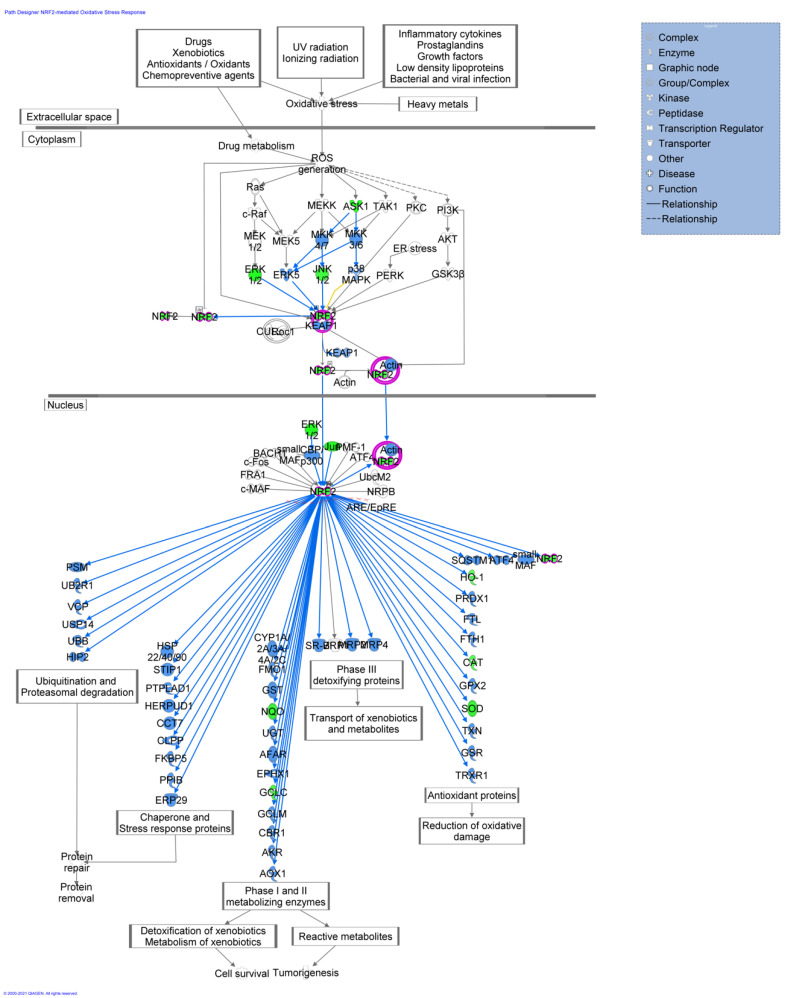
NRF2 oxidative stress signalling pathway was identified as a top canonical pathway by IPA following the analysis of gene changes in CCs cells from the smokers group when compared with the non-smokers group. Down-expressed genes are depicted in green while genes in white are the ones inferred by IPA.

**Figure 6 ijms-22-13147-f006:**
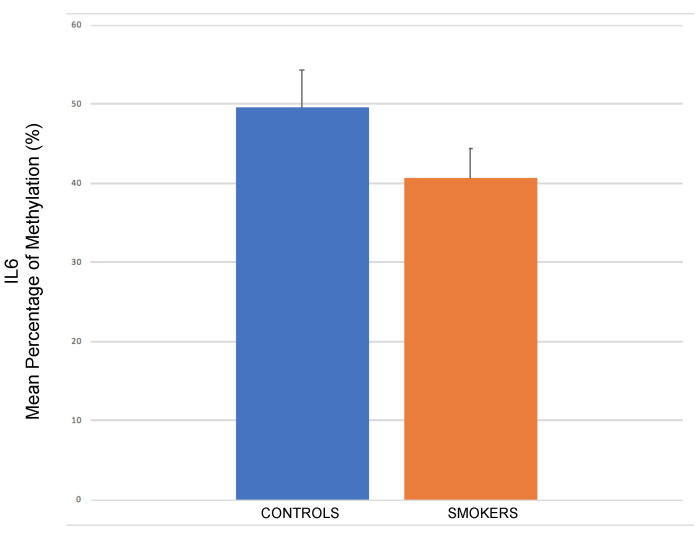
Significant mean down-methylation of IL-6 gene promoter in cumulus cells of female smokers compared to corresponding controls (*p* < 0.05).

**Table 1 ijms-22-13147-t001:** Patients’ demographic characteristics.

	Non Smokers (*n* = 5)	Smokers (*n* = 5)	*p*-Value
Mean n° of cigarettes smoked/daily	-	8.2	
Age	30.8 ± 0.8	33.2 ± 2.6	0.08
AMH	8.6 ± 2.5	9.2 ± 8.3	0.9
BMI	25.3 ± 4.5	24.6 ± 5.4	0.8
Previous IVF attempts	1.4 ± 0.5	1.4 ± 0.5	1
Days of stimulation	9.4 ± 2.1	9.8 ± 0.8	0.7
Total dosage of gonadotropin	1260.0 ± 697.6	1695.0 ± 1336.8	0.5
Oocytes retrieved at ovum pick-up	11.6 ± 2.1	12.2 ± 8.9	0.9
N° of MII oocytes	8.4 ± 1.9	8.8 ± 7.4	0.9

AMH: Anti-mullerian hormone, BMI: body mass index, IVF: in vitro fertilization, MII: meta phase II.

**Table 2 ijms-22-13147-t002:** List of under- and over-expressed genes following exposure to cigarette smoke in cumulus cells of smokers compared to controls, type(s) and cell location.

ID	Expr Log Ratio	Entrez Gene Name	Location	Type(s)
APEX1	−0.534	apurinic/apyrimidinic endodeoxyribonuclease 1	Nucleus	enzyme
BCL2	−0.528	BCL2 apoptosis regulator	Cytoplasm	transporter
CAT	−0.422	catalase	Cytoplasm	enzyme
CYBA	−0.414	cytochrome b-245 alpha chain	Cytoplasm	enzyme
EGFR	−0.480	epidermal growth factor receptor	Plasma Membrane	kinase
F3	−0.540	coagulation factor III, tissue factor	Plasma Membrane	transmembrane receptor
FOXO1	−0.435	forkhead box O1	Nucleus	transcription regulator
GCLC	−0.461	glutamate-cysteine ligase catalytic subunit	Cytoplasm	enzyme
GPX3	−0.416	glutathione peroxidase 3	Extracellular Space	enzyme
HIF1A	−0.539	hypoxia inducible factor 1 subunit alpha	Nucleus	transcription regulator
HMOX1	−0.408	heme oxygenase 1	Cytoplasm	enzyme
IL1B	−0.335	interleukin 1 beta	Extracellular Space	cytokine
IL6	2.432	interleukin 6	Extracellular Space	cytokine
JUN	−0.548	Jun proto-oncogene, AP-1 transcription factor subunit	Nucleus	transcription regulator
MAP3K5	−0.533	mitogen-activated protein kinase kinase kinase 5	Cytoplasm	kinase
MAPK1	−0.536	mitogen-activated protein kinase 1	Cytoplasm	kinase
MAPK8	−0.469	mitogen-activated protein kinase 8	Cytoplasm	kinase
MSRA	−0.452	methionine sulfoxide reductase A	Cytoplasm	enzyme
NFE2L2	−0.445	nuclear factor, erythroid 2 like 2	Nucleus	transcription regulator
NFKB1	−0.386	nuclear factor kappa B subunit 1	Nucleus	transcription regulator
NQO1	−0.495	NAD(P)H quinone dehydrogenase 1	Cytoplasm	enzyme
SERPINE1	−0.402	serpin family E member 1	Extracellular Space	other
SIRT1	−0.544	sirtuin 1	Nucleus	transcription regulator
SOD2	1.520	superoxide dismutase 2	Cytoplasm	enzyme
STAT1	−0.513	signal transducer and activator of transcription 1	Nucleus	transcription regulator
TP53	−0.364	tumor protein p53	Nucleus	transcription regulator

## Data Availability

The raw data associated with this study are available from the corresponding author upon reasonable request.

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
