# Peer review of "Impact of Cigarette Smoking on the Expression of Oxidative Stress-Related Genes in Cumulus Cells Retrieved from Healthy Women Undergoing IVF"

_ijms, 2021, doi:10.3390/ijms222313147_

Round 1
Reviewer 1 Report
The reviewer’s comments are as below:
- The research study is based on the expression of oxidative stress-related genes in human cumulus cells of female smokers undergoing IVF techniques. Overall, the manuscript is written very well and the analysis is in line with the test results to justify the objective of the proposed research study.
- Page 3 of 22, Table 1 – In the first row “Mean no. of cigarettes smoked/die”, Please clarify if it is smoked/daily or not. Please rephrase the terminology.
- There are some basic questions to clarify it further. For example, is there any specific type of cigarette or tobacco, will there be any impact on the test results depending on the type of tobacco, what about regular diets of these sample donors, will there be any impact if the diet contains high protein or an antioxidant component. These are not necessary information for this research study, however, it would be a great addition. The intention of asking this question is to include all the assumptions/limitations made here for this research study.
- The research study is mainly focused on the samples from healthy women undergoing IVF. What would be the author’s views on IVM techniques? Is there any possibility to correlate these two different techniques based on this research study?
- The discussion section explains very well about Gene expression theory and functional and network analysis, etc. however it is missing the root cause for these findings. What are the main components in tobacco or in cigarettes which influence these outcomes? It would be a good discussion if it can highlight the key toxic component for these affecting parameters.
Author Response
Q1. “Page 3 of 22, Table 1 – In the first row “Mean no. of cigarettes smoked/die”, Please clarify if it is smoked/daily or not. Please rephrase the terminology.”
A1. We made the required correction.
Q2. “There are some basic questions to clarify it further. For example, is there any specific type of cigarette or tobacco, will there be any impact on the test results depending on the type of tobacco, what about regular diets of these sample donors, will there be any impact if the diet contains high protein or an antioxidant component. These are not necessary information for this research study, however, it would be a great addition. The intention of asking this question is to include all the assumptions/limitations made here for this research study.”
A2. Thank you for the important suggestion. We included in our study patients smoking approximately the same number of cigarettes/daily for the same time, as highlighted in Table 1 and Paragraph 4.2 and patients with a BMI that is within the normal range (Table 1). In addition, we excluded from the study patients taking micronutrient supplementation (Paragraph 4.2). However, we are aware that the type of cigarettes and diet may represent relevant factors to be investigated and that the lack of this type of information in our study represents a limitation. To better clarify this aspect, we specified in the Materials and Methods section the exclusion criteria. In addition, we included these limitations in the text.
Q3. “The research study is mainly focused on the samples from healthy women undergoing IVF. What would be the author’s views on IVM techniques? Is there any possibility to correlate these two different techniques based on this research study?”
A3. This highlighted aspect is particularly important. In fact, ART includes various processes, from mild interventions like hormonal treatment to more invasive procedures like in vitro fertilization (IVF) or intracytoplasmic sperm injection (ICSI). Ovarian stimulation, in vitro maturation, fertilization, embryo culture, embryo biopsy, transfer, and cryopreservation all have the potential to modulate gene expression and epigenetic reprogramming and imprinting in gametes and early embryos. However, in our study we analysed CCs, following ovum pick-up, from smoking and non-smoking women undergoing the same ovarian stimulation and IVF protocols (ICSI), without the IVM step. Consequently, no correlation can be made between different techniques. We better specified it in the inclusion criteria.
Q4. “The discussion section explains very well about Gene expression theory and functional and network analysis, etc. however it is missing the root cause for these findings. What are the main components in tobacco or in cigarettes which influence these outcomes? It would be a good discussion if it can highlight the key toxic component for these affecting parameters.”
A4. Thank you for the relevant point. It results difficult to identify the main constituents that influenced the outcomes presented in our analysis considering that patients were exposed to cigarette smoke and not to its single components. Cigarette smoke is a mixture of about 4000 chemical compounds, however, several experimental studies conducted in vitro or in vivo showed that nicotine (with its principal metabolite- cotinine) together with benzo[a]pyrene (BaP), dimethylbenz[a]anthracene (DMBA) and cadmium exert central roles in the process of ovotoxicity influencing the steroidogenesis, the correct follicle development and the correct meiotic progression of the oocytes. Relevant information was included in the Discussion section.
Reviewer 2 Report
In this manuscript, the authors discussed the impact of smoking on the expression of oxidative stress-related genes in cumulus cells (CCs) in women undergoing IVF. The major finding of the manuscript is that there are 26 differently regulated genes in smokers in comparison with non-smokers undergoing IVF. These genes are involved in a protective cellular response to oxidative stress-caused damages. Additionally, the authors showed that the methylation level of IL6 is reduced that can explain the up-regulation of this gene.
Major concerns:
- At least half of the manuscript is a literature review on a function of mentioned 26 genes and links between some of them and NRF2. Instead of writing such an extended review, it will be more scientifically interesting to check if methylation levels of all 26 genes (not only IL6) are in line with idea that differences in methylation level between smokers and non-smokers. Basically, I would suggest studying a deeper biological perspective on why these genes are differentially regulated in smokers.
- Line 225-227. Did the authors perform any tests of CCs vitality or proliferation to confirm correlation with the discussed in the manuscript differentially regulated genes in smokers?
Minor concerns:
- Figure 1. Missing label for Y-axis.
- Figure 2. Please, increase the font size.
- Figure 4. Hard to read and digest. Please, increase the font size and highlight how the NRF2 partway is altering in the control vs CS group.
- Figure 6. In the cap should be “Significant mean down-regulation of…”.
In conclusion, there are interesting observations on the important topic in the manuscript but the story is too raw to be published. I would recommend reconsidering the manuscript after a major revision.
Author Response
Major concerns:
Q1. At least half of the manuscript is a literature review on a function of mentioned 26 genes and links between some of them and NRF2. Instead of writing such an extended review, it will be more scientifically interesting to check if methylation levels of all 26 genes (not only IL6) are in line with idea that differences in methylation level between smokers and non-smokers. Basically, I would suggest studying a deeper biological perspective on why these genes are differentially regulated in smokers.
A1. We thank the reviewer for raising this important point. Changes caused by epigenetic modifications of DNA represent dynamic information that could be markedly influenced by environmental stimuli, such as cigarette smoking. Although a number of studies have explored the mechanisms behind cigarette smoke-induced ovarian toxicity, several aspects related to this process remain, so far, not completely elucidated. For this reason, we focused our attention to evaluate, for the first time, the impact of cigarette smoking on the expression of oxidative stress-related genes in human cumulus cells of female smokers undergoing IVF techniques compared to age-matched non-smoking controls. Our scientific interest is centered around damage induced by cigarette smoke and reproduction. Gene expression of cumulus cells represent a well-known marker of oocyte competence, and the overall observed downregulation suggests a lower antioxidant capacity in CCs of smoking versus non-smoking women that could reflect the oocyte status. The gene function analysis of the modulated genes provides a support to our results explaining the gene expression theory and functional and network analysis. This evidence contributes to furtherly comprehend the molecular basis of female reproduction, a way to identify a gene expression signature as non-invasive biomarker and consider the importance of antioxidant treatments for smoking women undergoing IVF protocols. In conclusion, we completely agree with the observation of the referee, who however addressed a different important topic about the mechanisms able to regulate the gene expression in CCs. In a future study, it would be an interesting idea to unravel these processes, such as miRNA profile and methylation status in CCs of smokers vs non-smokers with a starting point a global epigenomic approach.
Q2. “Line 225-227. Did the authors perform any tests of CCs vitality or proliferation to confirm correlation with the discussed in the manuscript differentially regulated genes in smokers?”
A2. We thank the referee for this interesting observation. We used all retrieved CCs for gene expression analysis and did not perform any test of CCs vitality or proliferation.
Minor concerns:
Q3. Figure 1. Missing label for Y-axis.
A3. We apologize for this oversight. Added the missing label for Y-axis.
Q4. Figure 2. Please, increase the font size.
A4. Done
Q5. Figure 4. Hard to read and digest. Please, increase the font size and highlight how the NRF2 partway is altering in the control vs CS group.
A5. In line with this observation, we better explained the figure related to the NRF2 pathway (Figure 5). Font size has been increased and a description legend regarding all components and relationships present in the pathway has been inserted. The figure’s cap has also been adjusted in order to clarify IPA prediction analysis that shows an inhibition of the oxidative stress defense machinery. In addition, we better described the physiological function of the NRF2 pathway in the Results section.
Q6. Figure 6. In the cap should be “Significant mean down-regulation of…”.
A6. Following the referee’s suggestion the cap of Figure 6 has been modified, highlighting the detected down-methylation of the IL-6 gene promoter in cumulus cells of female smokers compared to their non-smoking controls.
Round 2
Reviewer 2 Report
I would suggest focusing on the deeper study of the biological side of this project. Results of this in-deep study would be very interesting and useful to improve IVF.